# Mothers’ and Children’s Mental Distress and Family Strain during the COVID-19 Pandemic: A Prospective Cohort Study

**DOI:** 10.3390/children10111725

**Published:** 2023-10-24

**Authors:** Janelle Boram Lee, Kharah M. Ross, Henry Ntanda, Kirsten M. Fiest, Nicole Letourneau

**Affiliations:** 1Department of Community Health Sciences, Cumming School of Medicine, University of Calgary, Calgary, AB T2N 1N4, Canada; janelle.lee1@ucalgary.ca (J.B.L.); kmfiest@ucalgary.ca (K.M.F.); 2Owerko Centre, Alberta Children’s Hospital Research Institute, University of Calgary, Calgary, AB T3B 2X9, Canada; henry.ntanda@ucalgary.ca; 3Department of Psychology, Centre for Social Sciences, Athabasca University, Athabasca, AB T9S 3A3, Canada; kharahr@athabascau.ca; 4Department of Critical Care Medicine, Cumming School of Medicine, University of Calgary, Calgary, AB T2N 1N4, Canada; 5Department of Psychiatry, Cumming School of Medicine, University of Calgary, Calgary, AB T2N 1N4, Canada; 6Faculty of Nursing, Cumming School of Medicine, University of Calgary, Calgary, AB T2N 1N4, Canada; 7Department of Paediatrics, Cumming School of Medicine, University of Calgary, Calgary, AB T2N 1N4, Canada

**Keywords:** family strain, maternal mental distress, child mental distress, COVID-19 pandemic, survey, longitudinal study, APrON study

## Abstract

Background: The COVID-19 pandemic had a widespread impact on families with dependent children. To better understand the impact of the pandemic on families’ health and relationships, we examined the association between mothers’ and children’s mental distress and family strain. Methods: Three waves of the COVID-19 Impact Survey were analyzed, collected from a subsample of mother–child pairs (*n* = 157) from the Alberta Pregnancy Outcomes and Nutrition (APrON) longitudinal cohort in Alberta, Canada. Latent class analyses were performed to determine patterns and group memberships in mothers’ and children’s mental distress and family strain. Multivariable logistic regression models were conducted to test associations between mothers’ and children’s mental distress and family strain trajectory classes. Results: Mothers with medium/high levels of mental distress were at increased odds of experiencing high family strain compared to those with low levels of distress (medium aOR = 3.90 [95% CI: 1.08–14.03]; high aOR = 4.57 [95% CI: 1.03–20.25]). The association between children’s mental distress and family strain was not significant (aOR = 1.75 [95% CI: 0.56–5.20]). Conclusion: Mothers’ mental distress, but not children’s, was associated with family strain during the pandemic. More distressed individuals experienced greater family strain over time, suggesting that this association may become a chronic problem.

## 1. Introduction

In March 2020, the SARS-CoV-2 (COVID-19) pandemic was declared by the World Health Organization (WHO). Worldwide, regulations limiting the coronavirus spread impacted individuals’ mental distress and family strain [1,2,3,4,5,6,7,8,9,10]. In particular, families with dependent children were adversely affected by struggles imposed by public health measures (e.g., school/childcare closure) [11,12], often straining family relationships [6,7,8,10]. Understanding these adverse family experiences is vital as society addresses COVID-19-related family challenges and moves forward to recovery from the consequences of the pandemic.

Family strain is variably defined to include quality of parenting, child abuse potential, caregiver burden and parenting distress [8,12,13]. The COVID-19 pandemic influenced overall family relationships, particularly caregiver strain, relationship strain and parenting-related conflicts. In a Canadian sample of 570 caregivers (97.9% mothers), over 75% reported moderate-to-high levels of caregiver strain during the pandemic [7]. In another Canadian study, 37.2% of 568 participants (predominantly mothers) experienced relationship distress [8]. Parents also reported increased parental psychological distress and parenting irritability, decreased family positive expressiveness, conflict with children and parenting-related exhaustion [6,14,15]. On the other hand, some parents reported that stay-at-home orders during the COVID-19 pandemic enhanced their relationship with children [14,16].

The COVID-19 pandemic had a widespread influence on parental and child mental health. In Western countries, parents and caregivers reported that their mental health and children’s behaviors had been negatively impacted since the pandemic [1,5,14,17,18,19]. Compared to pre-pandemic data, the rates of depression and anxiety in mothers of children across all ages increased during the COVID-19 pandemic [1]. Parents also experienced negative changes in mood and stress, potentially impacting parent–child relationships [12]. Furthermore, caregivers reported increased behavioral issues (e.g., tantrums, crying, sleep changes and decreased child talking) in their children [20]. Younger parental age, financial stress, previous parental and child physical and mental health conditions and COVID-19-related stressors contributed to more family strain during the pandemic than pre-pandemic data [6]. Caregivers with younger children, compared to those with older children, reported an increased level of mental distress and more family strain [6,7,21]. Furthermore, the significant association between parental anxiety and depression symptoms and parental stress during the pandemic was also exhibited [12]. COVID-19-related stressors and maternal mental distress were associated with higher child abuse potential, caregiver burden, low quality of parenting and parenting distress [8,12,13]. The pre-pandemic and COVID-19 literature suggested sociodemographic factors (e.g., household income and parental employment status) potentially confounding the link between family members’ mental health and family relationships [22,23,24,25].

Research examining family strain during the pandemic has often been limited to examining relationships between parent–partners or parent–child relationships rather than a more comprehensive definition examining relationships between different types of family members (e.g., siblings and adult partners with children). Further, to our knowledge, specific associations between mothers’ and children’s mental distress and family strain outcomes during the COVID-19 pandemic have not been described. Lastly, there is a lack of longitudinal evidence examining this specific link between mothers’ and children’s mental distress and family strain in the context of the COVID-19 pandemic. The contribution of longitudinal evidence is pertinent in understanding individual differences associated with various psychosocial risk factors [26].

The current prospective cohort study addressed the evidence gaps with novel approaches, using a more comprehensive definition of family strain, examining the particular associations between mothers’ and children’s mental distress and family strain in COVID-19 and employing longitudinal evidence. In this study, we sought to determine whether (1) mothers’ mental distress (i.e., depression, anxiety and perceived stress) increased the odds of family strain during the COVID-19 pandemic and (2) children’s mental distress (i.e., emotional, behavioral and social distress) increased the odds of family strain during the COVID-19 pandemic. We hypothesized that mothers’ and children’s mental distress would be associated with increased odds of experiencing family strain.

## 2. Materials and Methods

### 2.1. Study Design and Setting

This prospective cohort study was drawn from the Alberta Pregnancy Outcomes and Nutrition (APrON) COVID-19 Maternal Impact Survey designed to assess the effects of COVID-19 on families living in Alberta, Canada [27]. Three survey waves (approximately six months apart) were undertaken from May 2020 to July 2021: Wave 1—May to July 2020, Wave 2—November 2020 to January 2021 and Wave 3—May to July 2021. At the time of data collection, children of the APrON cohort ranged from 7 to 11 years old. A recent systematic review found that the mental health of children under 12 years old has been negatively impacted during the pandemic; the most at-risk children were those with increased pandemic-related stressors and worsened mental health [28]. Details on the APrON Study and the COVID-19 Maternal Impact Survey are published elsewhere [27,29].

### 2.2. Participants

Mothers were invited to complete the baseline COVID-19 Maternal Impact Survey within four months of March 15, 2020, when the Provincial Chief Medical Officer of Health implemented school closure and physical distancing measures to prevent the spread of COVID-19 [27]. In the COVID-19 Maternal Impact Survey, 639 mothers participated via an online platform, REDCap. For this study, we included mothers who responded to all three survey waves, *n* = 173 (at Wave 1, *n* = 358 responded; at Wave 2, *n* = 358 responded; at Wave 3, *n* = 329 responded). Out of 173 respondents, 157 mothers completed the family strain questionnaire.

### 2.3. Variables and Measures

#### 2.3.1. Family Strain

Family strain during the COVID-19 pandemic was measured in the three waves. Mothers reported whether the pandemic strained their family relationships, including the respondent with her partner, the respondent with her child in the APrON study and the respondent’s partner’s relationship with their child in the APrON study. On a three-point Likert scale, each question ranged from 1 (It has brought us closer together), 2 (Not much has changed) to 3 (It has strained our relationship), with 4 indicating Not Applicable; Cronbach’s α was from 0.62 to 0.66 in the current survey data. At each wave, the outcome (family strain) dichotomous variable (yes/no) was derived as follows: anyone who responded as “It has strained our relationship” was coded “Yes” and “It has brought us closer together” or “Not much has changed” was coded “No”. “NA” respondents at each time point were excluded. The outcome was coded as ‘Yes’ for those with a ‘Yes’ at any of the waves and ‘No’ for those with a ‘No’ consistently at the three waves.

#### 2.3.2. Mothers’ Mental Distress

Mothers’ mental distress was assessed through measures of depression, anxiety and stress at each assessment. The Centre for Epidemiologic Studies Depression Scale (CES-D) was administered to measure depressive symptomatology [30]. The CES-D scale is a ten-item self-report measure on a four-point Likert scale. This scale asks participants to rate their experiences of depressive symptoms in the past week: 0 (Rarely or none of the time (less than 1 day)), 1 (Some or a little of the time (1–2 days)), 2 (Occasionally or more moderate amount of the time (3–4 days)) and 3 (More or all of the time (5–7 days)). The total score was obtained by reverse scoring negatively worded responses and then summing 10 items ranging from 0 to 30. Higher scores indicated higher levels of distress. In the current survey data, Cronbach’s *α* ranged from 0.82 to 0.83.

Mothers’ anxiety was self-reported using the State-Trait Anxiety Inventory (STAI) questionnaire [31]. State anxiety is “subjective, consciously perceived feelings of tension and apprehension and heightened autonomic nervous system activity” [31]. This scale consists of twenty items rated on a four-point: 1 (Not at all), 2 (Somewhat), 3 (Moderately so) and 4 (Very much so), with Cronbach’s *α* = 0.83 [32]. A total score was obtained by reverse scoring negatively worded responses and then summing up all the items. The total score ranged from 6–24, with higher scores representing a higher state of anxiety.

The Perceived Stress Scale (PSS) [33] was used to measure maternal perception of stress. PSS is a 10-item scale that measures how individuals perceive their stressful life situations, with an internal consistency of *α* = 0.86 [34]. Each of the items is rated on a five-point Likert scale, which ranges from 0 (Never), 1 (Almost Never), 2 (Sometimes), 3 (Fairly Often) to 4 (Very Often). The total PSS score was obtained by reverse scoring responses to the four positively stated items and then summing across all scale items. The total stress scores ranged from 0 to 40, with higher scores indicating more perceived stress.

Each depression, anxiety and stress variable was standardized separately at each wave (observed response score − average score/standard deviation). A mother’s mental distress variable was created by averaging these three standardized variables at each wave.

#### 2.3.3. Children’s Mental Distress

Children’s mental distress was calculated as a total difficulties score employing the Strength and Difficulties Questionnaire (SDQ) [35]. SDQ is a globally recognized instrument for assessing the mental health status of children and young people aged 4–17. The SDQ is a twenty-five-item scale questionnaire self-reported with five subscales by mothers: emotional symptoms (items 3, 8, 13, 16 and 24), conduct problem (5, 7, 12, 18 and 22), hyperactivity (items 2, 10, 15, 21 and 25), prosocial (items 1, 4, 9, 17 and 20) and peer problem (items 6, 11, 14, 19 and 23) scales. The items are rated on a three-point Likert scale: 0 (Not true), 1(Somewhat true) and 2 (Certainly true). For each of the five subscales, the total scores ranged from 0 to 10 and were calculated if at least three items had been completed; otherwise, the summary score was considered missing. Negatively worded items (7, 11, 14, 21 and 25) were reverse-scored before computing the total score for each subscale. The total SDQ score is a summation of the emotional, conduct, hyperactivity and peer problem subscales; the prosocial scale was excluded as this was computed at each wave. The total score ranged from 0 to 40. Higher scores represented higher distress in children, excluding the prosocial subscale.

#### 2.3.4. Covariates

Due to variability in the socioeconomic status (SES) variable in the APrON cohort, we obtained SES scores by summing the values of maternal income, education and marital status based on the following dichotomies [36,37]: income < $70,000 (0) or ≥ $70,000 (1), educational attainment of <a university degree (0) or ≥a university degree (1) and not married (0) or married (1). The sample’s SES scores ranged from 0–3, with higher scores indicating a higher SES level.

The following COVID-19-related covariates were included in the final logistic regression models: worries about child well-being (dichotomous [low/high]), worries about child education (dichotomous [low/high]), family difficulty managing school work and childcare (dichotomous [low/high]), and family difficulty managing child’s school activities (dichotomous [low/high]). Mothers were given a set of eight questions to rate about supporting their children’s well-being and education experiences during the pandemic. Five questions about their children’s well-being included: “I am concerned about my child’s behavioural challenges, outbursts or short temper”, “I am worried that my child is sad or depressed”, “I am worried that my child is anxious”, “I feel that my child is currently receiving adequate amounts of physical activity” and “I feel that my child is currently receiving adequate amounts of sleep”. Mothers also responded to the following three questions about their child’s education: “I feel that my child will be academically ready for the next school year”, “I feel that my child will re-adjust socially (reconnecting with or making new friends) for the next school year” and “I feel that my child is able to keep up with his/her schoolwork”. Each question was rated on a five-point Likert scale: 1 (Strongly agree), 2 (Agree), 3 (Neither agree nor disagree), 4 (Disagree) and 5 (Strongly disagree). Summing all items computed at each wave, the total well-being score ranged from 1 to 25, and the total education score ranged from 1 to 15. Higher scores represented better well-being and worse educational experiences in children.

### 2.4. Statistical Analysis

Descriptive statistics were presented to summarize the sample demographic characteristics. To identify the trajectories of mothers’ mental distress, children’s mental distress and family strain across the three waves, latent class analyses were conducted for each variable using MPlus version 7.11 [38]. A one-class model was fit first, followed by fitting successive models with more classes in order to identify the most parsimonious models. Model solutions were evaluated by comparing likelihood ratio statistics (L2), the Akaike information criterion (AIC) and the Bayesian information criterion (BIC) across successive models. Better-fitting models had lower L2, AIC and BIC values. Entropy, an index for assessing the precision of assigning latent class membership, evaluated further model solutions. The Vuong–Lo–Mendell–Rubin likelihood ratio test (VLMR-L2) was used to test statistically significant differences between the models based on *p* < 0.05.

The following multivariable logistic regression models were constructed to assess the relationships between (1) the trajectory class of mothers’ mental distress (exposure) and family strain (outcome) and (2) the trajectory class of children’s mental distress (exposure) and family strain (outcome). Models were adjusted for covariates and presented as odds ratios (ORs) and 95% confidence intervals (CIs). The presence of effect modification in the models was assessed by adding interaction terms and testing for their statistical significance (*p* < 0.05). Confounding effects were assessed by employing stepwise methods; if covariates were *α* < 0.25 (i.e., worries about child well-being and child education) and if *α* ≥ 0.25 but clinically relevant (i.e., sociodemographic variable), they were retained in the final models.

## 3. Results

### 3.1. Latent Class Analysis Trajectories

#### 3.1.1. Mothers’ Mental Distress

Latent class models specifying one to four models were estimated, and the three-class model was accepted as the final model (Appendix A; Table A1). The three-class model fit indexes (measured with L2, BIC, AIC and VLMR-L2) were lower than the one- and two-class models, indicating an improved fit. The VLMR-L2 showed a significant difference between the two- and three-class models, suggesting that the three-class model gave a significant improvement in model fit. Differences between the three- and four-class models were not significant, suggesting that the four-class model was more parsimonious. The entropy value for the three-class model was high (0.819), indicating an acceptable precision in assigning individual cases to their appropriate class, and all latent classes for the three-class model had a sufficient sample size.

#### 3.1.2. Children’s Mental Distress

Latent class models specifying one to three models were estimated; the two-class model was accepted as the final model (Appendix A; Table A2). The two-class model fit indexes (L2, AIC and BIC) were lower than the one- and two-class models, indicating an improved fit. The VLMR-L2 demonstrated a significant difference between the two- and three-class models; this suggests a significant fit improvement in the two-class model. The three-class model was not selected as the difference between the one- and two-class models was not significant. The entropy value for the two-class model was high (0.819), suggesting acceptable precision in assigning individual cases to their appropriate class. The sample size of each latent class in the two-class model was found to be sufficient.

#### 3.1.3. COVID-19-Related Child Well-Being and Education Variables

We estimated one to three latent class models (Appendix A; Table A3 and Table A4). The model fit assessments (L2, AIC, BIC and VLMR-L2) indicated the two-class model as the final model for both variables. For the two-class model, the entropy value was high (well-being: 0.74, education: 0.97); this suggested acceptable precision in assigning individual cases to their appropriate class. Further examination found that the two-class model showed a sufficient sample size.

### 3.2. Descriptive Sample Characteristics

#### 3.2.1. Frequencies of Sample Characteristic Variables

Table 1 summarizes the sample characteristics of eligible study participants. Of the total participants (*N* = 157), 80.89% of mothers reported having experienced family strain (between respondent and partner, respondent and her child, respondent’s partner and child and child siblings) in their households during the COVID-19 pandemic. Mothers who experienced high depression, anxiety and/or stress during the pandemic comprised 19.11% of the total sample. Mothers reported that 20.38% of their children experienced high distress across the emotional, conduct, hyperactivity and peer problem scales. More than 70% of mothers reported being worried about their child’s well-being. The mean of the sociodemographic scores (e.g., maternal education, marital status and household income) was 2.71 (standard deviation (SD) = 0.58, range: 0–3).

#### 3.2.2. Frequencies of Family Strain by Mothers’ and Children’s Mental Distress Trajectory Class

Table 2 summarizes the frequencies of the family strain outcome by mothers’ and children’s mental distress status. Frequencies of family strain increased in proportion amongst each mother’s mental distress group from low to high levels (low: 6.90%, medium: 23.19%, high: 33.33%). This pattern was similar for children’s mental distress status (low: 16.00%, high: 31.25%).

### 3.3. Multivariable Logistic Regression

Compared to the low-maternal-distress group, and adjusting for covariates, mothers with a medium level of mental distress (OR = 3.90, 95% CI: 1.08, 14.03) and mothers with high levels of mental distress (OR = 4.57, 95% CI: 1.30, 20.25; Table 3) were also more likely to experience family strain. The models also examined children’s mental distress trajectory classes and family strain outcome. Compared to the low-child-mental-distress trajectory class, adjusting for covariates, children in the high-distress class did not significantly experience differences in family strain (OR = 1.75, 95% CI (0.56, 5.20); Table 3).

## 4. Discussion

This prospective cohort study examined associations between mothers’ and children’s mental distress trajectory classes and family strain during the COVID-19 pandemic. The study findings demonstrated a significant association between mothers’ mental distress trajectory class and the odds of experiencing family strain over the follow-up period, but not between child mental distress and family strain. This study highlights that when mothers struggle with higher levels of distress, regardless of whether it is due to depression, anxiety or stress, households are more likely to experience family strain. We found that more mentally distressed mothers experienced more significant family strain over time, suggesting this association as a potentially chronic problem.

The current study found a significant association between maternal mental distress and increased odds of experiencing family strain during the COVID-19 pandemic, which was consistent with the pre-pandemic literature. Before the pandemic, the family burden was related to personal mental health problems of family members [39]. Another study demonstrated that caregiver burden was associated with caregivers’ psychological distress; however, family functioning did not influence caregiver strain or psychological distress separately [40]. The current study using data during the COVID-19 pandemic found no statistically significant association between child mental distress and family strain. There are inconsistencies in the pre-pandemic literature examining this specific relationship. Children’s mental health was associated with parental stress but not parental involvement with their child [41]. Also, a reciprocal relationship was examined between parental relationship quality and children’s externalizing and internalizing problems [42]. Aligning with the current study findings, a recent Chinese study examined the associations between youth mental health, parental mental health distress and family relationships during the COVID-19 outbreak and found that parent–child and marital relationships are associated with parental mental health [43]. However, Bai et al. (2022) used a sample of families with children aged from 10 to 18. In contrast, the current study focused on families with children aged from 7 to 11 years due to the relevancy of pre-pandemic evidence that a younger age contributed to more-strained family relationships [6]. Given the current study finding aligning with the pre-pandemic literature, we speculate the long-term ramifications on family strain to be linked with mothers’ and children’s mental health distress beyond the pandemic context. Thus, continued attention should be given to supporting the impacted families.

To our knowledge, this is the first North American study to examine the association between mothers’ and children’s mental distress and family strain during the COVID-19 pandemic. Past studies have examined only parental strains or proxies of family strains (e.g., quality of parenting, child abuse potential, caregiver burden and parenting distress) in association with maternal mental health and child distress in the context of the COVID-19 pandemic [8,12,13,44] and have ignored children’s contributions to family strain (or its converse higher family functioning). A recent European study found that decreases in parental strain were associated with decreases in child problem behaviors and increases in child well-being during the pandemic [44], which differs from the specific relationships examined in this current study, where we did not find a significant relationship between child mental distress and family strain. This study used a comprehensive definition of family strain; family strain was examined across different types of family members in a household (i.e., adult partners, child siblings and adult partners with kid[s]). This aggregated definition is a novel approach; it suggests that a higher level of maternal mental health distress is associated with mothers’ strained relationships with other family members and their relationships with each other. Thus, this finding suggests a broader implication for considering all family members when tailoring support to improve family relationships impacted by maternal mental health distress.

The limitations of this study include potential selection biases in the sample. The current study lacked variability in SES variables (i.e., maternal education, marital status and household income). We used a composite SES score to address these variables’ lack of variability. Also, the study sample was skewed towards high SES values (e.g., sociodemographic variable score average = 2.71; range 0 to 3). As the literature suggests that a higher SES value has protective effects on family strain and mental health in parents and children, we speculate that the current finding underestimates the true relationship [45,46,47,48,49]. Due to the nature of self-reported data, this study may also be susceptible to recall bias, leading to potential misclassification of the exposure and outcome variables. Furthermore, even though this study examined the associations between maternal and child mental distress and family strain at different time points, we cannot infer a causality of the relationship in this observational study.

## 5. Conclusions

The current research demonstrated that maternal mental distress was associated with family relationships between partners, caregivers and children and child siblings during the COVID-19 pandemic. As the pandemic impacted parents and children at a whole-population level, we recognize its impact on population health and urge scholars to explore potential long-term risks associated with family strain and psychological distress. At policy levels, we recommend that increased attention be paid to tangible resources to develop supportive environments for children and families, such as quality childcare, safe neighbourhoods and expanded parental leave policies. Furthermore, policymakers should prioritize developing public health interventions to mitigate the consequences of mental health distress on family relationships beyond the pandemic-related impacts. We also recommend that clinicians expand treatments from the individual level to the family and community level when working with this impacted population, as mental health distress is linked with various types of family strain. Policymakers and clinicians should collaborate to provide an integrated system of longer-term and family-level support services to improve family relationships and maternal–child distress.

## Figures and Tables

**Table 1 children-10-01725-t001:** Frequency of demographic and COVID-19-related variables of participants (*N* = 157).

	**Frequency**	**Percentage**
Family strain during the COVID-19 pandemic		
Yes	30	80.89
No	127	19.11
Mothers’ mental distress		
Low	58	36.94
Medium	69	43.95
High	30	19.11
Children’s mental distress		
Low	125	79.62
High	32	20.38
Worries about child’s well-being		
Low	46	29.30
High	111	70.70
Worries about child’s education		
Low	132	84.08
High	25	15.92
	**Mean (SD ** ^ **1** ^ **)**	**Range**
Family difficulty managing schoolwork and childcare (at Wave 1)	7.15 (2.46)	3–15
Family difficulty with school activities (at Wave 1)	7.20 (2.47)	0–15
Family difficulty with school activities (at Wave 2)	6.27 (2.45)	0–13
Socioeconomic status (SES) score ^2^	2.71 (0.58)	0–3

^1^ Standard deviation; ^2^ SES score was calculated from maternal income, education and marital status by summing the values according to the following dichotomies: income < $70,000 (0) or ≥ $70,000 (1), educational attainment of <a university degree (0) or ≥a university degree (1) and not married (0) or married (1).

**Table 2 children-10-01725-t002:** Frequencies of family strain by mothers’ mental distress status class (low, medium, vs. high) and children’s mental distress class (low vs. high) during the COVID-19 pandemic (*N* = 157).

Family Strain	Mothers’ Mental Distress, *n* (%)			
	Low	Medium		High
Experienced family strain during the COVID-19 pandemic	4 (6.90)	16 (23.19)		10 (33.33)
Did not experience family strain during the COVID-19 pandemic	54 (93.10)	53 (76.81)		20 (66.67)
	**Children’s Mental Distress,** * ** n** * ** (%)**			
	**Low**		**High**	
Experienced family strain during the COVID-19 pandemic	20 (16.00)		10 (31.25)	
Did not experience family strain during the COVID-19 pandemic	105 (84.00)		22 (68.75)	

**Table 3 children-10-01725-t003:** Unadjusted and adjusted odds ratios of family strain by mothers’ and children’s mental distress status classes during the COVID-19 pandemic (*N* = 157).

	Family Strain	
	Unadjusted OR ^1^ (95% CI ^2^)	Adjusted OR ^1^ (95% CI ^2^)
Mothers’ mental distress		
Low	Referent	Referent
Medium	4.08 (1.28, 12.99)	3.90 (1.08, 14.03) ^3^
High	6.75 (1.90, 23.99)	4.57 (1.03, 20.25) ^3^
Children’s mental distress		
Low	Referent	Referent
High	2.39 (0.98, 5.80)	1.75 (0.56, 5.20) ^4^

^1^ Odds ratio; ^2^ confidence interval; ^3^ adjusted for worries about child well-being, worries about child education, family difficulty managing schoolwork and childcare, difficulty managing child’s school activities and sociodemographic score (education, marital status and household income); ^4^ adjusted for maternal mental health distress, family difficulty managing schoolwork and childcare, difficulty managing child’s school activities and sociodemographic score (education, marital status and household income).

## Data Availability

Data available on request due to restrictions e.g., privacy or ethical. The data presented in this study are available on request from the corresponding author.

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
