# Peer review of "Mothers’ and Children’s Mental Distress and Family Strain during the COVID-19 Pandemic: A Prospective Cohort Study"

_children, 2023, doi:10.3390/children10111725_

Round 1

Reviewer 1 Report

Dear Authors,

I have had the opportunity to review your manuscript and I would like to provide some constructive feedback based on my assessment. Firstly, let me congratulate you for the meticulous execution of your research, which reflects a strong methodological foundation and comprehensive statistical analysis. It is evident that your study was conducted with precision and rigor, which is commendable. While the research is scientifically sound, it appears to lack novelty, and the narrative does not captivate the reader's interest as much as it could. To enhance the manuscript's appeal consider highlight novelty. The manuscript could benefit from a more engaging writing style. Beyond the immediate context of pandemic times, it would be valuable to discuss the transferable implications of your research. How can your findings inform policy decisions, public health strategies, or broader societal considerations? Be explicit in suggesting real-world applications and potential policy recommendations. When discussing the implications of your research, clearly communicate the significance of your findings and how they can drive positive change. Make sure your readers understand the broader implications and transformative potential of your work. I look forward to seeing how these suggestions will further improve your manuscript.

Regards.

Author Response

Dear Reviewer,

We sincerely appreciate your review and feedback on our manuscript titled “Mothers’ and Children’s Mental Distress and Family Strain During the COVID-19 Pandemic: A Prospective Cohort Study.”

We added the discussion of potential transferable implications of the research beyond the pandemic context and expanded policy decisions, public health strategies, or broader societal considerations in the Discussion and Conclusion sections. Please review:

Given the current study finding aligning with the pre-pandemic literature, we speculate the long-term ramifications on family strain linked with mothers’ and children’s mental health distress beyond the pandemic context. Thus, continued attention should be given to supporting the impacted families.” (lines 317-321);

At policy levels, we recommend increased attention to tangible resources to develop supportive environments for children and families, such as quality childcare, safe neighbourhoods and expanded parental leave policies. Furthermore, policymakers should prioritize developing public health interventions to mitigate the consequences of mental health distress on family relationships beyond the pandemic-related impacts. We also recommend that clinicians expand treatments from the individual level to the family- and community level when working with this impacted population, as the im-pact of mental health distress is linked with various types of family strain. Policymakers and clinicians should collaborate to provide an integrated system of longer-term and family-level support services to improve family relationships and maternal-child distress.” (lines 357-367)

We also highlighted the novelty of this work on lines 86-90:

This prospective cohort study addressed the evidence gaps with novel approaches: using a more comprehensive definitions of family strain, examining the particular as-sociations between mothers’ and children’s mental distress and family strain in the COVID-19 context and employing longitudinal evidence.

Furthermore, as recommended by Ms. Elva Zhang, Assistant Editor, we have revised the language throughout the manuscript. Our objective was to mitigate any instances of repetition, as identified in the iThenticate report.

Should you have any additional comments or suggestions, please reach out to me via email (Janelle.lee1@ucalgary.ca). We sincerely appreciate your valuable feedback and time.  

Best regards,

Janelle Boram Lee

Reviewer 2 Report

Dear authors,

Your manuscript Mothers’ and Children’s Mental Distress and Family Strain During the COVID-19 Pandemic: A Prospective Cohort Study, is very well formed and written and very interested. I only suggest a few little things.

-          I suggest that you state earlier in the description of the respondents why you decided on the age of the children of 7-11 years. Although you stated the explanation in the discussion part, however, I consider it useful that you clarify it earlier.

-          Can you explain how the control of participants was carried out in all three waves of the pandemic. How did you connect the respondents' answers and know that it was the same respondents in all three waves of the pandemic?

-          the instruments used in the research are described very nicely and in detail, and I suggest that if you quote Cronbach, you do this for all instruments.

-          in line 337 the parenthesis should be closed

-          the listing of references should be aligned with the journal's instructions in the list as well as in the manuscript. I recommend that the period be written after the number as is customary.

Best regards,

Author Response

Dear Reviewer,

We sincerely appreciate your review and feedback on our manuscript titled “Mothers’ and Children’s Mental Distress and Family Strain During the COVID-19 Pandemic: A Prospective Cohort Study.”

We have addressed the following comments below. We sincerely appreciate your thorough review.

  • Mothers of children of 7-11 years were invited to participate in the baseline COVId-19 Maternal Impact Survey as part of the currently ongoing cohort from the Alberta Pregnancy Outcomes and Nutrition (APrON) longitudinal cohort in Alberta, Canada. This section was added to address the importance: “At the time of data collection, children of the APrON cohort ranged from 7 to 11 years old. A recent systematic review found that the mental health of children under 12 years old has been negatively impacted during the pandemic; most at-risk children were those with increased pandemic-related stressors and worsened mental health [28]. Details on the APrON Study and the COVID-19 Maternal Impact Survey are published elsewhere [27, 29].” (lines 100-107)
  • We confirm the reference groups were the same groups across the waves based on the latent class analysis, i.e., mothers’ mental distress (low) and children’s mental distress (low). Please see under 1. Latent Class Analysis Trajectories (line 224)
  • The Cronbach’s alphas for Family Strain and Mother’s Mental Distress were calculated based on the current survey data. We clarified by adding: “…in the current survey data” (line 124, 139).
  • For STAI and PSS measures, the references were added. (lines 145, 150).
  • In line 337, the parenthesis was closed.
  • The listing of references was edited to fit the journal’s instructions.

Furthermore, as recommended by Ms. Elva Zhang, Assistant Editor, we have revised the language throughout the manuscript. Our objective was to mitigate any instances of repetition, as identified in the iThenticate report.

Should you have any additional comments or suggestions, please reach out to me via email (Janelle.lee1@ucalgary.ca). We sincerely appreciate your valuable feedback and time. 

Best regards,

Janelle Boram Lee
